# An Improved Robust Method for Pose Estimation of Cylindrical Parts with Interference Features

**DOI:** 10.3390/s19102234

**Published:** 2019-05-14

**Authors:** Jieyu Zhang, Yuanying Qiu, Xuechao Duan, Kangli Xu, Changqi Yang

**Affiliations:** 1Key Laboratory of Electronic Equipment Structure Design of Ministry of Education, Xidian University, Xi’an 710071, China; zhangjieyu@stu.xidian.edu.cn (J.Z.); xchduan@xidian.edu.cn (X.D.); 2School of Mechano-Electronic Engineering, Xidian University, Xi’an 710071, China; klxu@stu.xidian.edu.cn; 3Shanghai Spaceflight Precision Machinery Institute, Shanghai 201600, China; yang_qi_2000@163.com

**Keywords:** pose measurement, laser scanning, ellipse fitting, interference feature, M-estimation, robustness

## Abstract

Horizontal docking assembly is a fundamental process in the aerospace assembly, where intelligent measurement and adjustable support systems are urgently needed to achieve higher automation and precision. Thus, a laser scanning approach is employed to obtain the point cloud from a laser scanning sensor. And a method of section profile fitting is put forward to solve the pose parameters from the data cloud acquired by the laser scanning sensor. Firstly, the data is segmented into planar profiles by a series of parallel planes, and ellipse fitting is employed to estimate each center of the section profiles. Secondly, the pose of the part can be obtained through a spatial straight line fitting with these profile centers. However, there may be some interference features on the surface of the parts in the practical assembly process, which will cause negative effects to the measurement. Aiming at the interferences, a robust method improved from M-estimation and RANSAC is proposed to enhance the measurement robustness. The proportion of the inner points in a whole profile point set is set as a judgment criterion to validate each planar profile. Finally, a prototype is fabricated, a series of experiments have been conducted to verify the proposed method.

## 1. Introduction

In the field of aerospace assembly and pipeline welding, it is usually necessary to dock one cylindrical part with another within tolerance [1]. In order to make the process more automatic and intelligent, a complete measurement system, control system, and execution system are required in assembly line design. Thus, measurement approaches for the cylindrical parts and the robustness in the actual environment should be discussed in depth at the very beginning because many problems exist in engineering, such as the complex on-site environment and interference features on the parts.

At present, the frequently used pose measurement in the automation field can be divided into target-based measurement and non-target measurement according to whether the measurement target or marker is used or not [2].

For target-based measurement, it is required to mount additional targets or spray markers on the surface of the object. The pose parameters can then be calculated using the fixed positional relationship between the target and the object after acquiring the position of the targets. For instance, in aircraft assembly, the laser tracker system (LTS) is usually employed to measure the spatial pose of large-sized parts such as aircraft cabins [3], and the cabins can be aligned through 3-DOF support systems [4]. The accuracy of the LTS is high enough, but it can only track one position of a single target at the same time with this approach. To obtain the pose of the part, it is essential to move the measurement target manually on the surface, which will decrease the measurement precision due to the manual operation [5]. Furthermore, the poor redundancy of the data leads to a low robustness in complex situations. Besides, the above process can also be accomplished by machine vision [6,7]. The spatial positions of the markers that are sprayed on the surface of the object are photographed by cameras, and then the pose parameters can be calculated from the relationship between the markers and the object. Manual operations with the target ball can be avoided by methods with markers, which will reduce the complexity of the operation. However, the measurement precision is limited by the precision of machine vision, which is factually inversely related to the measuring range.

In general, there are two major drawbacks in target-based measurement. One is the requirement of targets or markers, which will reduce the productivity and automation in industrial productions. The other one is that considerable pose errors will be introduced due to the inaccurate positional relationship between the targets and the parts.

One can notice that the existence of measurement targets is not conducive to the implementation of automated processes [8], so numerous studies are carried out to explore how to avoid measurement targets or markers, which can be named as non-target approaches.

Most of the non-target measurement methods are based on binocular vision or distributed monocular vision. These approaches can be divided into three categories, point matching [9], geometric primitive recognition [10,11], and model matching [12,13]. Among them, the point matching refers to a method of taking pictures of the same object through a binocular camera and calculating three-dimensional information by matching corresponding feature points on the two pictures acquired from cameras. Unfortunately, there are some uncertainties in the search process of matching point pairs, so the accuracy and robustness are poor in practical engineering [8]. In the second approach, geometric primitives such as lines, planes, and circles are extracted from the pictures acquired by cameras, and the pose can be estimated through the geometric relationship between the recognized geometric primitives and the object. This approach finds wide applications in micro assembly as a result of the clear measurement background. For example, in Liu’s research [14], two orthogonally arranged cameras are employed to take the pictures of the object, and the edges in the photographs are extracted to estimate the pose of the measured object. The model matching approach refers to comparing the projection of the existing CAD model with the image taken by the camera, and obtaining the posture by the ICP (Iterative Closest Points) or other similar algorithms. However, the existing CAD model is required in this method. Furthermore, it is difficult to improve the accuracy through this approach due to the difference between the actual machining size and the CAD model [15]. In summary, all three approaches based on machine vision have strict environmental requirements which may increase the complexity and cost in actual industry application.

Consequently, this paper presents a method of laser scanning to improve the measurement robustness that can be applied in practical engineering. However, the technology of laser scanning, or named structured light triangulation focus more on the measurement of the geometric shape of the object itself, such as welding detection [16,17,18], geomorphological detection [19], and three-dimensional reconstruction [20]. So far, however, only a small volume of work has been published about pose measurement for cylindrical parts by laser scanning. In approximate problems, Yu-cun Zhang et al. [21] proposed a method of structured light scanning for the geometric measurement of cylindrical forgings, but pose measurement is not discussed. Measurement accuracy for cylinders by laser scanning is discussed in the works of Rahayem et al. [22,23]. They evaluated the accuracy of ellipse fitting about the incomplete elliptical arcs obtained by laser scanning. These studies have been more concerned with geometry measurement, but pose measurement is less discussed. In the research of Turgut et al. [24], a laser scanning sensor mounted to a robot was employed to scan the specific rectangular and circular features on the part to obtain the posture. However, this method requires further machining of specific features on the surface of the object to be measured, which is usually unacceptable in actual engineering.

This paper aims to obtain the pose parameters of cylindrical parts precisely by the laser scanning approach. In this approach, the side surface of the part is scanned by a laser scanning sensor along a linear module to acquire the point cloud. The data is then segmented by a series of parallel planes into some planar elliptical profile point sets. Subsequently, the least square fitting is adopted to estimate the centers of each point set. A spatial straight line fitting is then applied on these centers to obtain the pose of the cylindrical part and the parts can be aligned by 4-DOF manipulators according to the pose parameters.

However, the surface of the object is not always an ideal cylindrical surface in actual engineering. This fact will introduce an adverse effect on the least square fitting of the ellipse centers. The causes of this problem are various, for example, the geometric interface features such as holes and bulges, the specular reflection of the laser, the occlusion of the fixture and the measurement noise, etc. Therefore, the above approaches are far from enough to get the pose parameters precisely, corresponding robust enhancements must be adopted to improve the reliability of the system. According to this demand, commonly used approaches involve M-estimation [25], least square median filtering [26], and random sampling consistency (RANSAC) [27]. For the robust fitting problem of the ellipse, Halir et al. [25] proposed an estimation method based on M-estimation, which can obtain the ellipse center more accurately in the case of external point interference. However, one disadvantage is that the algorithm may fail when the amount of external points is large. For RANSAC, Chen et al. [28] identified obscured pupils by RANSAC, through which the correct position of the pupil center can be obtained with a high probability, but the calculation efficiency is relatively low. In general, the M-estimation always has a smaller computing load and RANSAC approach has stronger robustness. Thus, we improve the M-estimation to valid the profile sets and eliminate the invalid sets.

From the above studies, it can be inferred that few studies tackle with the non-target measurement for cylindrical parts in the premise of accuracy and robustness. For this reason, a synthesis method which can be applied in practical production is proposed in this paper for the automatic assembly line of cylindrical parts. A robust enhancement derived from M-estimation and RANSAC is carried out to deal with the interferences in a complex industrial environment. Moreover, a prototype is designed and fabricated to verify the effectiveness of the proposed robust method.

The content of this paper is arranged as follows: In Section 2 the measurement principle and algorithm are introduced. In Section 3, an improved M-estimation approach is proposed to avoid various interference features in actual assembly. In Section 4, a prototype is designed and fabricated to verify the proposed method. And the full text is summarized in Section 5.

## 2. Experimental Scheme and Measurement Principle

### 2.1. Axis Pose Measurement Method Based on Profile Scanning

This paper designs a method based on structured light triangulation. Meanwhile, a precise adjustment based on two 4-DOF manipulators is designed to meet the requirement of the precise alignment of the cylindrical parts.

The measurement-adjustment system is illustrated in Figure 1. As shown in Figure 1a, a 6-dimensional vector ${\left(X_{C0},Y_{C0},Z_{C0},\alpha ,\beta ,\gamma \right)}^{T}$ is adopted to describe the pose of the cylindrical part, where ${\left(X_{C0},Y_{C0},Z_{C0}\right)}^{T}$ is coordinate of the center of end surface $P_{C0}$, α is the rotation angle about the axis, β, γ are the projection angles of the axis to the coordinate plane $X_{0}O_{0}Y_{0}$ and $X_{0}O_{0}Z_{0}$, i.e., deflection angle β and pitch angle γ.

As illustrated in Figure 1b, two 4-DOF manipulators are mounted on the assembly rail to support the cylindrical part. Each manipulator has three degrees of freedom in the orthogonal Cartesian direction and one degree of freedom of rotation about the axis of the fixture. When the part is clamped by the two manipulators, the pose error can be compensated in six degrees of freedom so that to dock one part to the other precisely. Therefore, precise measurement is essential to guide the assembly process. In the system shown in Figure 1b, a laser scanning sensor is employed to scan the side surface of the part and five pose parameters of the six, that is ${\left(X_{C0},Y_{C0},Z_{C0},\beta ,\gamma \right)}^{T}$ can be gained. According to these parameters, two cylindrical parts are aligned completely and mating features on two mating surfaces are photographed by two industrial cameras, then the relative angle Δα is determined and adjusted.

The laser scanning sensor used in this paper is a two-dimensional laser ranging sensor based on the line structured light triangulation, it is thus also called the laser profile scanner [19], or the LVS (Laser Vision Scanner). The core technology, i.e., structured light triangulation, has been advanced for more than 30 years [29]. During this period, there have been many practical discussions on engineering application, but most of them focus on the measurement of the geometric morphology of the parts, such as the measurement of the geometric shape of parts [16,30], weld detection [5,18]. In recent years, the combination of the line structured light triangulation method and robotic technology has further promoted the automation of measurement. Although few studies have explored how to measure the pose of parts by laser scanning, high accuracy, and robustness of the laser measurement endow it a strong application prospect in precision assembly fields.

The measurement principle of laser scanning is shown in Figure 2. A laser projector emits a laser strip onto the measured parts, and then the strip was captured by the industrial camera. The spatial position of each point in the laser strip can then be gained according to the encoder and mapping relationship that is determined by the prior calibration.

During the measurement process, the laser scanning sensor moves along the direction parallel to the $X_{0}$ axis. This process can be regarded as to obtain the corresponding profile point sets $L_{i}$ by intersecting a series of parallel sections with the surface of the cylinder.

According to the space geometry, each profile on the cross-section belongs to an elliptic arc with the same radius *R*. The arc is defined by the following equations:(1)$$\left\{\begin{array}{l} a=R\sqrt{1+\tan^{2}\beta +\tan^{2}\gamma }\\ b=R\\ \theta =\arctan(\tan\beta /\tan\gamma ) \end{array}\right.,$$
where a, b are the lengths of the semi-major and semi-minor axis respectively, θ is the rotation angle of the ellipse. It can be inferred from Equation (1) that the shapes of the ellipses on each intersection are determined by the pose angles β, γ and the radius R of the cylinder together.

As shown in Figure 3, parameters of the ellipses can be obtained by ellipse estimation from the profile sets $L_{i}$ and then the center $P_{Ci}\left(y_{Ci},z_{Ci}\right)$
(i=1,2,⋯,N) of each ellipse namely axis-fitting points can be calculated. A straight line fitting is then carried out to obtain the pose parameters of the axis from the axis-fitting points.

### 2.2. Algorithms for the Centers of the Ellipses on Each Intersection

There are several algorithms for the ellipse estimation from profile sets $L_{i}$, such as the Hough transformation and least square fitting. Among them, Hough transformation is widely used in machine vision. It can be used to recognize multiple ellipses at the same time, but lags far behind the least square method in calculation accuracy and efficiency. Considering that there is only one ellipse corresponding to each set of profile points, the least square fitting is more suitable for the situation in this paper.

The least square fitting can be divided into two categories, i.e., the algebraic approach [31] and the geometric approach [32], in which the objective function is either the algebraic distance or geometric distance, respectively.

For most general ellipse fitting problems, the fitting points are sparsely distributed on the whole ellipses. In this situation, the former approach has lower accuracy but faster speed, while the latter is on the opposite. However, the fitting points acquired by laser scanning are distributed on one side of the ellipse densely and redundantly [33], so further discussion of the fitting precision and the computing load of the algorithms is necessary.

Rahayem Mohamed et al. [22] evaluated the comprehensive performances of algebraic and geometric methods utilizing the laser scanning data, and considered that the algebraic ellipse fitting proposed by Fitzgibbon et al. achieves a better balance in accuracy and efficiency.

For the M fitting points, the objective function is,
(2)$$\underset{a}{\min}\sum_{j=1}^{M}F\left(a,x_{j}\right)$$
where $F\left(a,x_{j}\right)=0$ is the equation of the ellipse, i.e.,
(3)$$F\left(a,x_{j}\right)=x_{j}\cdot a=Ax_{j}^{2}+Bx_{j}y_{j}+Cy_{j}^{2}+Dx_{j}+Ey_{j}+F=0,$$
where $a={\left[\begin{array}{llllll} A & B & C & D & E & F \end{array}\right]}^{T}$ is the parameter vector of the elliptic equation and $x_{j}=\left[\begin{array}{llllll} x_{j}^{2} & x_{j}y_{j} & y_{j}^{2} & x_{j} & y_{j} & 1 \end{array}\right]$ is a design vector. Meanwhile, inequality constraint $B^{2}-4AC<0$ is attached to Equation (3) to make it an ellipse function. In this algorithm, it can be replaced by an equality constraint:(4)$$B^{2}-4AC=-1,$$
which can be written as matrix form,
(5)$$a^{T}Ca=-1.$$

The matrix C is
(6)$$C=\left[\begin{array}{llll} 0 & 0 & -2 & \\ 0 & 1 & 0 & 0_{3\times 3}\\ -2 & 0 & 0 & \\ & 0_{3\times 3} & & 0_{3\times 3} \end{array}\right].$$

Thus, the objective function in Equation (2) can be rewritten as,
(7)$$\underset{a}{\min}{\left\|Da\right\|}^{2},$$
where D is a n×6 design matrix, that is $D=[\begin{array}{llll} x_{1} & x_{2} & \cdots & x_{M} \end{array}]$.

Solving Equation (5) together with Equation (7) by Lagrange multiplier, the parameter vector a can be obtained and the corresponding center $\left(y_{Ci},z_{Ci}\right)$ of the point set $L_{i}$ can be obtained by the following equations, where N is the amount of the profile sets.
(8)$$\left\{\begin{array}{c} y_{Ci}=\frac{BE-2CD}{4AC-B^{2}}\\ z_{Ci}=\frac{BD-2AE}{4AC-B^{2}} \end{array}\right.(i=1,2,\cdots ,N).$$

## 3. Robust Enhancement Algorithm for Pose Estimation

### 3.1. Ellipse Fitting

Theoretically, the above pose estimation method can obtain the pose of the axis from the profile data of an ideal cylindrical part. However, there are often various interference features on the surfaces of the actual workpiece, such as holes or bulges, which will result in a failed measurement. Therefore, the robustness of the fitting algorithm must be taken into consideration.

From the previous chapter, it can be learned that the basis of the proposed method for pose measurement is to segment point cloud into 2D profile sets $L_{i}$. An effective culling strategy should be put forward to select the profiles with less disturbance, i.e., valid profiles, and remove the profiles with more disturbances named invalid profile. In addition, the points belonging to the cylindrical surface are named inner points and those not belonging to the cylindrical surface are named outer points. In this way, the validity of profile point sets can be defined by the proportion of the inner points in a whole point set. That case indicates, when the proportion is more than the given threshold $p_{inner}$, the profile is considered valid, otherwise, it is invalid. In this paper, the threshold $p_{inner}$ is defined as 85%.

As shown in Figure 4, there are complex interferences in the actual point cloud that is acquired from the cylindrical part. Four typical profile point sets corresponding to the cross-sections A, B, C, and D in Figure 4 are shown in Figure 5.

For the above four cases in Figure 5, it is very difficult for the profile points in the last three cases to acquire the centers correctly using existing fitting approaches, such as the algebraic method [31] and the geometric method [32]. To estimate the centers as precisely as possible, there are two alternative enhancement methods, namely RANSAC, M-estimation, and its improvement [34,35]. M-estimation refers to repeatedly fitting and assigning different weights to each fitting point according to the residual until the weight of the inner points approaches one, while the weight of the outside points approaches zero. Its improvement, Msplit estimation, can detect multiple circles robustly in one point set [35]. The basic idea of RANSAC is to randomly select a small subset of data points, then fit them and calculate how many inner points are matched to the fitting model, and iterate the process until there is a greater probability to find an appropriate fitting model.

For the requirement of continuous fitting for ellipses, M-estimation is fast and suitable when the amount of the inside points is large. However, when the proportion of outer points is large, correct results can hardly be obtained. Conversely, RANSAC can better cope with the situation of fewer inside points. Unfortunately, its calculation cost is much more expensive. Thus, this paper improves the M-estimation by adding some characteristics from RANSAC approach, which makes it possible to validate the profiles by setting a threshold for the proportion of the inner points.

The main steps of the improved M-estimation are as follows:

(1) The fit model $F\left(a,x_{j}\right)$ is set as Equation (3), the object function can be written as
(9)$$f(a)=\sum_{j=1}^{M}\omega_{j}{\left[F\left(a,x_{j}\right)\right]}^{2},$$
where a is the parameter vector, $x_{i}$ is the coordinate of data points, M is the number of the fitting points, $\omega_{j}$ is the weight, which the default is $\omega_{j}=1$. For ellipse fitting, the parameter vector a can be solved according to Fitzgibbon’s research [31].

(2) By substituting a into $F\left(a,x_{j}\right)$, the model $F_{a}\left(x_{j}\right)$ for data point $x_{j}$ can be obtained. And the algebraic residual can be calculated as follows:(10)$$r_{j}=F_{a}\left(x_{j}\right).$$

(3) A small constant ε should be given as an error threshold. The data points $x_{j}$ satisfying $\left|r_{j}\right|<\varepsilon$ are picked out and named inner points. The proportion p of the inner points in all data points can then be obtained. If p>85%, the corresponding fitting data set is considered valid.

(4) Calculate the weight $\omega_{j}$ of each fitting point by the Tukey weight [25] to minimize the influence of outer points on fitting, i.e.,
(11)$$\omega_{j}\left(r_{j}\right)=\left\{\begin{array}{lll} {\left[1-{\left(\left|r_{j}\right|/c\sigma \right)}^{2}\right]}^{2} & \;\mathrm{if}\; & \left|r_{j}\right|<c\sigma \\ 0 & \;\mathrm{otherwise} \end{array}\right.,$$
where c=4.6851 is the tuning coefficient, which maximizes the efficiency of the M-estimation, σ is the robust standard deviation which can be calculated, that is,
(12)$$\sigma =1.4826\;\underset{j}{\mathrm{median}}\left|r_{j}\right|,$$
where median(·) is the median operator.

(5) Repeat the step (1)–(5) for T times. Terminate the cycling process and judge the validity. If the profile is valid in step (3), record the parameter vector a and calculate the center by Equation (8). Experiments show that accurate results can be obtained when the upper limit N of circulation is 3–5.

In order to evaluate the effectiveness of the above method, a numerical simulation is conducted with the profiles approximating the actual profiles in Figure 5. Assuming that the radius of the part R=200 mm, two pose angles $\beta =5^{\circ }$ and $\gamma =5^{\circ }$, it can be inferred that the semi-major axis a of the ellipse on each cross-section is 201.4 mm, the semi-minor axis b is 200 mm, and the rotation angle θ is $45^{\circ }$ according to Equation (1). The theoretical center of each profile is set to coincide with the point (0,0) in order for the convenience of analysis, and the amount of the fitting points is set as 640. The real proportion of the inner points is given in Table 1. Gaussian noise of standard deviation σ=0.05 mm is applied to all points along the horizontal and vertical directions to simulate the measurement error of the sensor. Fitting these data using the proposed approach and non-robust approach, the results are shown in Figure 6.

Four situations in Figure 5 are simulated by constructed data shown in Figure 6, these are, (a) smooth profile; (b) profile through the screw holes; (c) profile through the straight holes; (d) profile through the fixture. Centers of the four kinds of profiles estimated by two approaches are listed in Table 1 and Table 2, respectively.

It can be seen in Table 1 and Table 2 that the accuracy of the estimated center from Figure 6b has a great improvement with the proposed method and meets accuracy requirements of 0.05 mm. The improved results that are achieved are shown in Figure 6c. Since the proportion of the inner points cannot satisfy the accuracy and validity requirement, the profile data in this area are abandoned. Profile point set in Figure 6d is discarded in this method because of the small proportion of measured inner points. For the situations in Figure 6b–d, when the non-robust algebraic least square fitting is applied to estimate the centers of the profiles, the fitting model deviates from the true value due to the outer points, and the residual of all data points is too large, thus most of the points are judged as outer points. However, the improved M-estimation can effectively eliminate the outer points in Figure 6b and increase the proportion of inner points from 7.6% to 93.2%, exceeding the set threshold of 85%. Similarly, the proportion of inner points in Figure 6c is increased from 2.1% to 81.3%, nearly reaching the set threshold.

To sum up, the proposed method weakens the bad influences caused by the interference features, and very clearly improves the feasibility of pose measurement in practical engineering.

### 3.2. Ellipse Fitting

After the estimation for N point sets $L_{i}$, the centers of all valid profiles are recorded as axis fitting points $P_{Ci}\left(x_{ci},y_{ci},z_{ci}\right)$, i=1,2,⋯,N. A spatial straight line fitting is then conducted to solve the pose of the axis.

Assume that the axis of a cylindrical part intersects the plane $Y_{0}O_{0}Z_{0}$ at $P_{0}(0,y_{C0},z_{C0})$, the equation can be expressed as
(13)$$\left\{\begin{array}{c} y-px-y_{0}=0\\ z-qx-z_{0}=0 \end{array}\right.,$$
where $T={\left(1,p,q\right)}^{T}$ is the direction vector of the axis. Thus, the objective function can be written as
(14)$$\left\{\begin{array}{c} Q_{y}(p,y_{0})=\omega_{i}\sum_{i=1}^{N}{\left(y_{i}-px_{i}-y_{0}\right)}^{2}\\ Q_{z}(q,z_{0})=\omega_{i}\sum_{i=1}^{N}{\left(z_{i}-qx_{i}-z_{0}\right)}^{2} \end{array}\right..$$

Minimize two equations in Equation (14) separately, T and $P_{0}$ can be obtained.

The algebraic residual Equation (10) can be replaced by the geometric distance from a point to a straight line, i.e.,
(15)$$r_{i}=F_{a}\left(x_{i}\right)=\frac{P_{Ci}P_{0}\times T}{\left|T\right|}.$$

Furthermore, the pose of the axis can be described by $P_{0}$ and two pose angle β and γ, which can be obtained from the direction vector $T={\left(1,p,q\right)}^{T}$ by the following equations,
(16)$$\left\{\begin{array}{l} \beta =\tan^{-1}(p)\\ \gamma =\tan^{-1}(q) \end{array}\right..$$

Thus, the flow diagram of the proposed robust method for pose estimation can be given out in Figure 7.

## 4. Prototype Experiment

### 4.1. Design of the Prototype

To verify the effectiveness of the proposed method, we designed and fabricated an automatic assembly system.

As shown in Figure 8a, a commercial Gocator 2350 laser scanner manufactured by the LMI Technology based on the line structured light triangulation is mounted on a linear module which can drive the sensor scan the parts on the two assembly manipulators. A laser tracker system (LTS), shown in Figure 8b, is employed as a comparison.

Considering the error accumulation in the motion process of the mechanism, the measurement-adjustment process shown in Figure 9 is divided into the following three steps.

In step 1, one cylindrical part is set as the fixed target part and the other one is the adjusted one. The pitch and deflection angles (β,γ) of the adjusted part are acquired by the proposed robust approach. The part is then adjusted to horizontality. In step 2, the parts are scanned again and aligned accurately. In the final step, the pin and hole on the mating surfaces are photographed by two cameras and the part is rotated to complete the assembly.

Afterward, the validity of the proposed method and the non-robust method will be tested and compared by using the system mentioned above.

### 4.2. Design of the Prototype

The point cloud of a typical cylindrical part with interferences are acquired by the above system and drawn in Figure 4. The proposed method is conducted to solve for the pose parameters of the axis. Meanwhile, the other method without any enhancement of robustness is also carried out as a comparison [36].

The estimated centers of the profiles on each cross-section are fitted using the above two methods, and the variation of *Y* and *Z* coordinate with *X* axis are shown in Figure 10. Meanwhile, the side view of the cylindrical part is shown in Figure 11 as a reference, in which each area is labeled corresponding to Figure 10. In order to compare the final precision of both methods, the projection of the estimated axis and the projection of the axis measured by the LTS are also drawn in Figure 10.

From the side view in Figure 10 and Figure 11, it can be seen that there are straight holes in area A and D, occluded areas of the fixtures in area B and F, screw holes in area C and E. Thus, we can notice that the non-robust method fails to cope with any interference. In area A, large errors are introduced by the non-robust method due to the straight hole while the error data are abandoned by the proposed method. In area B, D, and F, the fitting centers obtained by the non-robust method are obviously wrong. However, they are effectively recognized and eliminated by the proposed method. In area E where there are some screw holes, the centers of on all the cross-sections are recognized due to less interference. In addition, the coordinates obtained by the non-robust method still fluctuate greatly, but these points are identified and discarded by the robust method in this paper. As a result, it can be seen from Figure 10 that the axis is successfully obtained by the proposed method, while it failed by the non-robust method.

The results in Figure 10 are quantified in Table 3. It can be concluded that in the case of interference, the non-robust method cannot get the correct result because it is not able to eliminate the interferences. The invalid profiles are discarded by the proposed robust method. The results show that the measured angle deviation between the prototype and the LTS is less than 0.02°, the position deviation of the axis is less than 0.3 mm. That is to say, the robust method proposed in this paper is proven to be effective from the engineering point of view.

### 4.3. Adjustment for the Cylindrical Part with Interference Features

In order to verify the measurement-adjustment system, an all-in-one alignment experiment is conducted. Firstly, the manipulator is adjusted to a general pose and the pose parameter are measured as $\beta =0.3575^{\circ }$, $\gamma =1.1127^{\circ }$, $y_{C0}=-8.272\;\mathrm{mm}$, $z_{C0}=6.514\;\mathrm{mm}$. A measurement-adjustment process is then conducted and the result is listed in Table 4. Moreover, the LTS is introduced to evaluate the error between the pose parameters acquired individually by the prototype and the LTS.

Table 4 shows that the pose of a cylindrical part with interference features can be precisely measured and aligned with the proposed system. After comparing the measured data of LTS with the target pose, it can be inferred that the angle adjustment accuracy is about 0.02 degrees and the position adjustment accuracy is less than 0.2 mm, which proves the validity of the proposed method for pose measurement.

## 5. Conclusions

Aiming at the high-accuracy pose measurement for cylindrical parts in industrial automated production line, this paper proposed a robust method for the pose estimation of the actual cylindrical parts that involve varieties of typical disturbances. An approach based on the M-estimation and RANSAC is applied to enhance the measurement robustness. Finally, a prototype is fabricated to verify the proposed robust measuring method. Some meaningful conclusions can be drawn as follows:

(1) The dense 3D point clouds acquired from a laser scanner are highly redundant which will reduce the measuring efficiency and precision. To address this problem, we proposed a fast and accurate algorithm solving the pose parameters from the point cloud. In this method, the data are segmented into 2D elliptical profile sets by a series of parallel cross-sections. A robust fitting is then conducted to estimate the centers of the profiles, further, a spatial straight line fitting is carried out to get the pose of the cylindrical axis.

(2) In order to make the algorithm feasible in practical engineering, a robust enhancement method is proposed for the interferences on the surface of the parts, such as screw holes, straight holes and fixture occlusion which may lead to a failure measurement. This method improved from the M-estimation and RANSAC can not only suppress the isolated noise points on each profile, but also filter the invalid profiles in the point cloud. In this method, the validity of profile point set is determined by the proportion of the inner points in the whole point set, i.e., if the proportion is greater than the given threshold $p_{inner}$, the profile is considered valid.

(3) A prototype is developed to evaluate the precision of the measuring method for pose estimation and effectiveness of the adjusting system. Additionally, a laser tracker system is introduced as a benchmark to validate the method of this paper. The result shows the angular and positional accuracy are 0.02° and 0.3 mm with the proposed method respectively, and the parts can be fitted together automatically and precisely through the adjustment of the prototype.

The method proposed in this paper is not confined to aerospace assembly. It can be widely applied in practical engineering to guide the assembly of cylindrical objects such as tunnels and silos. The robust data processing in this method can also handle the point cloud acquired by LiDAR (light detection and ranging) or other approaches for the similar cylindrical objects with interferences. In addition, the design of manipulators in the prototype which has been proved in a practical experiment can also supply references for the relevant research.

## Figures and Tables

**Figure 1 sensors-19-02234-f001:**
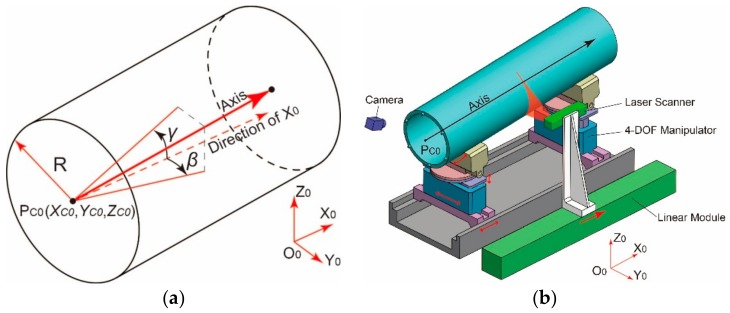
Diagram of the measurement and adjustable support system: (**a**) pose direction of the axis; (**b**) diagram of the structure.

**Figure 2 sensors-19-02234-f002:**
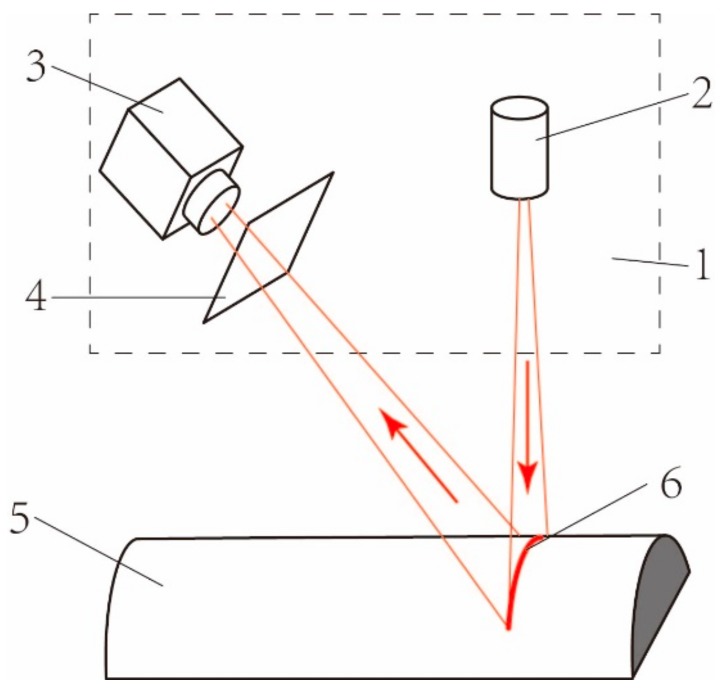
Diagram of a typical laser scanning sensor: 1. Laser scanning sensor; 2. Laser projector; 3. Camera; 4. Imaging Plane; 5. Part to be measured; 6. Laser strip.

**Figure 3 sensors-19-02234-f003:**
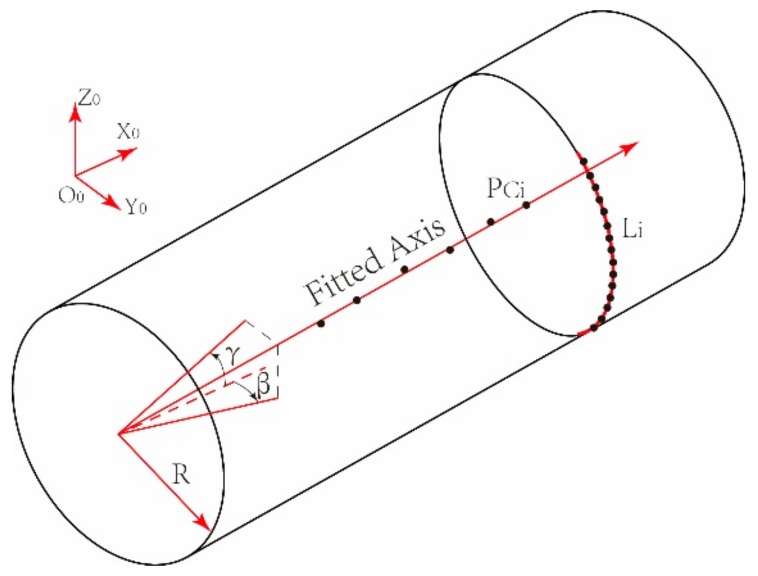
Diagram of posture estimation for the axis.

**Figure 4 sensors-19-02234-f004:**
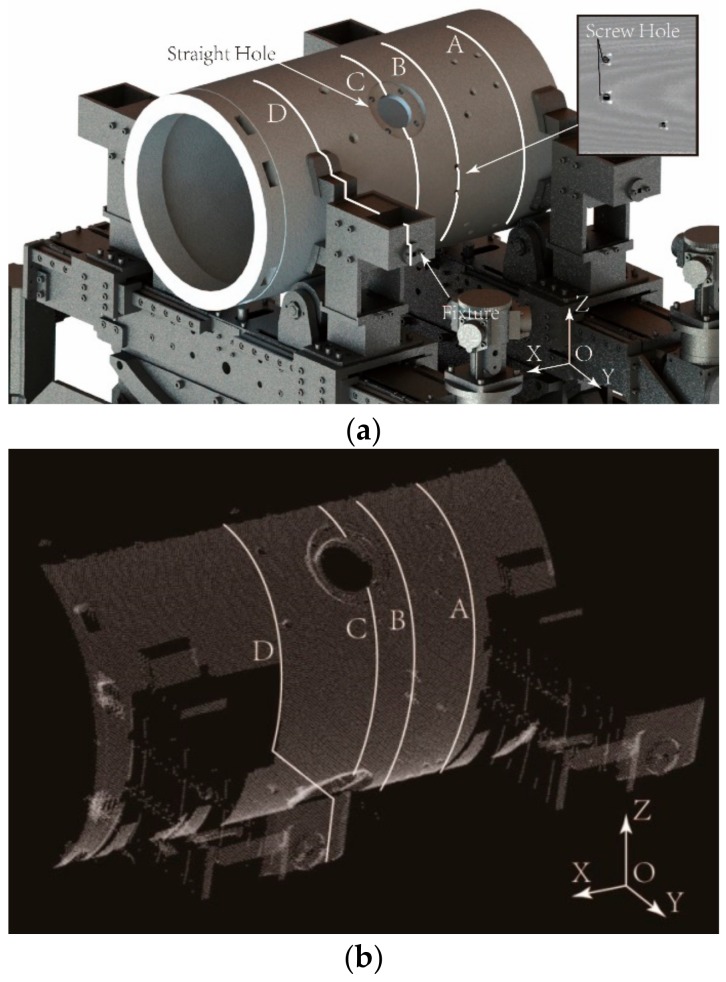
Measurement for the actual part with complex interferences: (**a**) diagram of the part on the support system; (**b**) point cloud obtained by the laser scanner.

**Figure 5 sensors-19-02234-f005:**
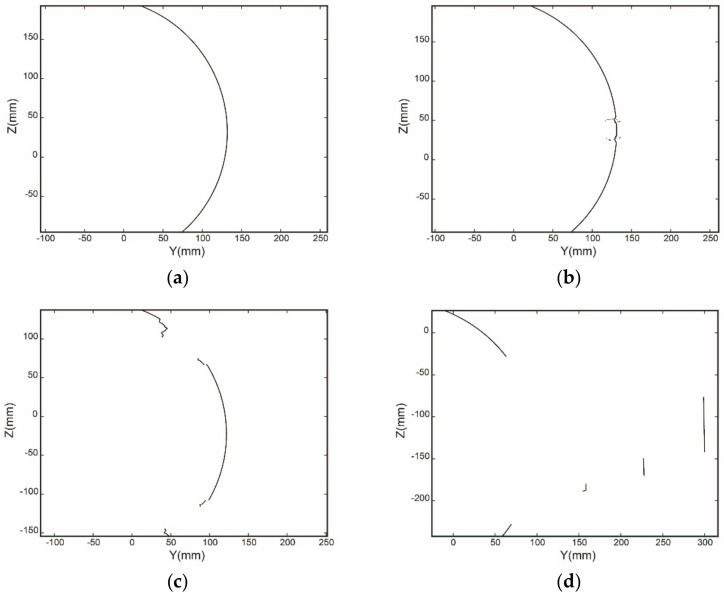
Four acquired ellipse profile point sets from different areas: (**a**) smooth profile; (**b**) profile through screw holes; (**c**) profile through straight hole; (**d**) profile through fixtures.

**Figure 6 sensors-19-02234-f006:**
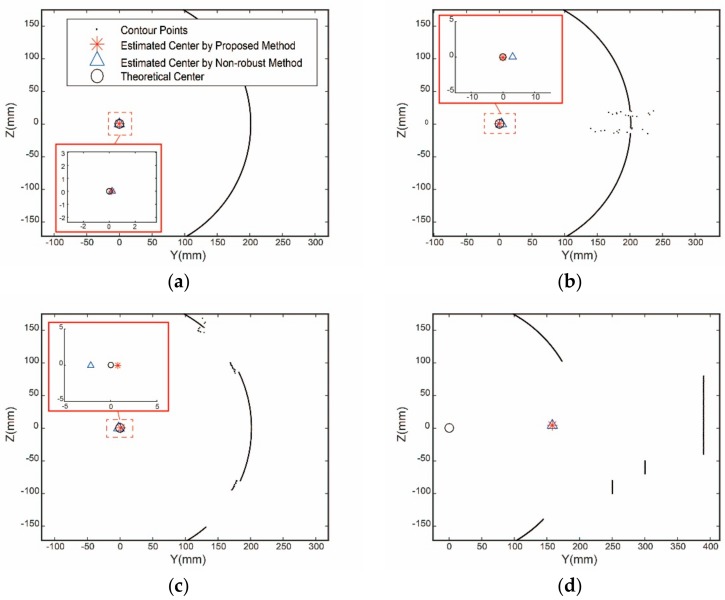
Constructed profile data: (**a**) constructed profile of area A; (**b**) constructed profile of area B; (**c**) constructed profile of area C; (**d**) constructed profile of area D.

**Figure 7 sensors-19-02234-f007:**
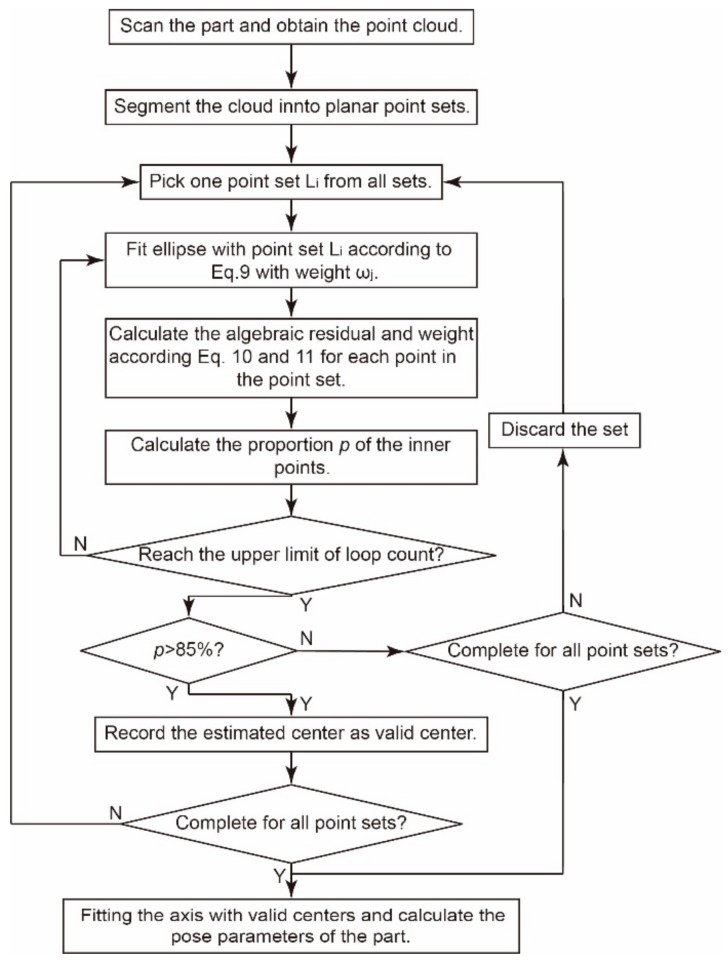
Flow diagram of the proposed robust method.

**Figure 8 sensors-19-02234-f008:**
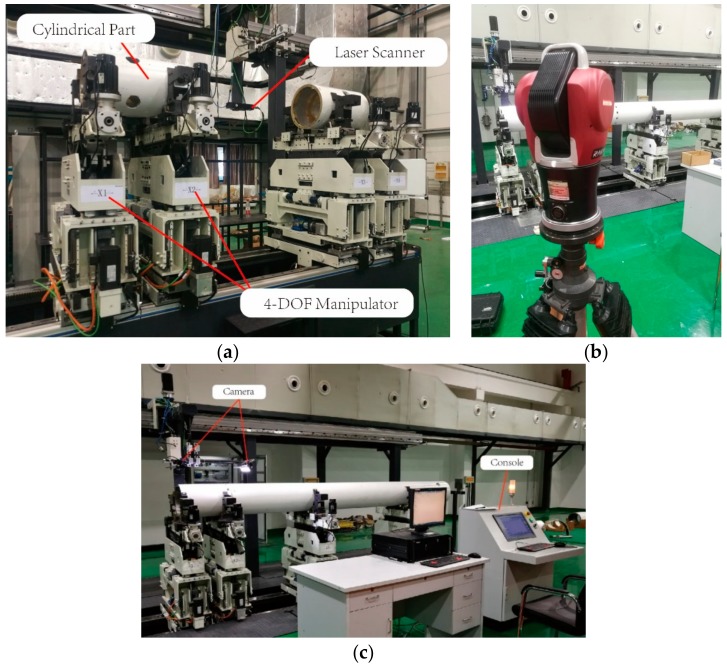
Prototype: (**a**) automatic assembly system; (**b**) laser tracker system(LTS); (**c**) prototype after an assembly processing.

**Figure 9 sensors-19-02234-f009:**
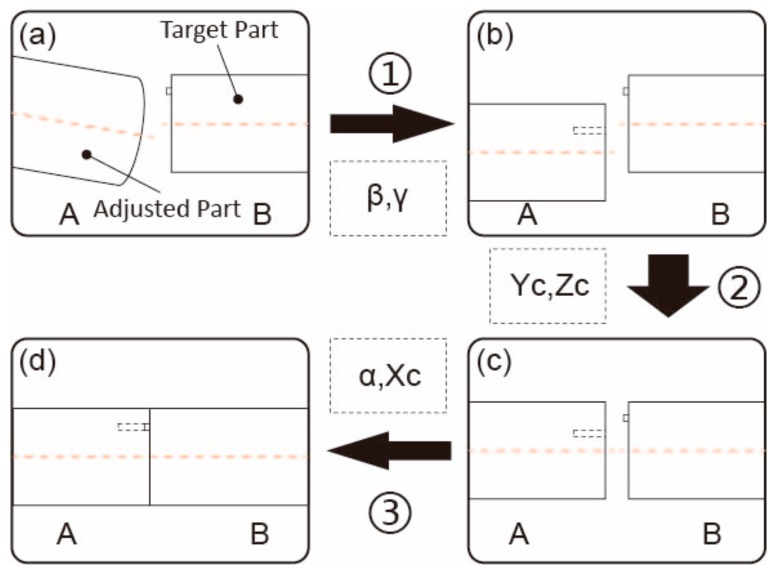
Measurement-adjustment Process.

**Figure 10 sensors-19-02234-f010:**
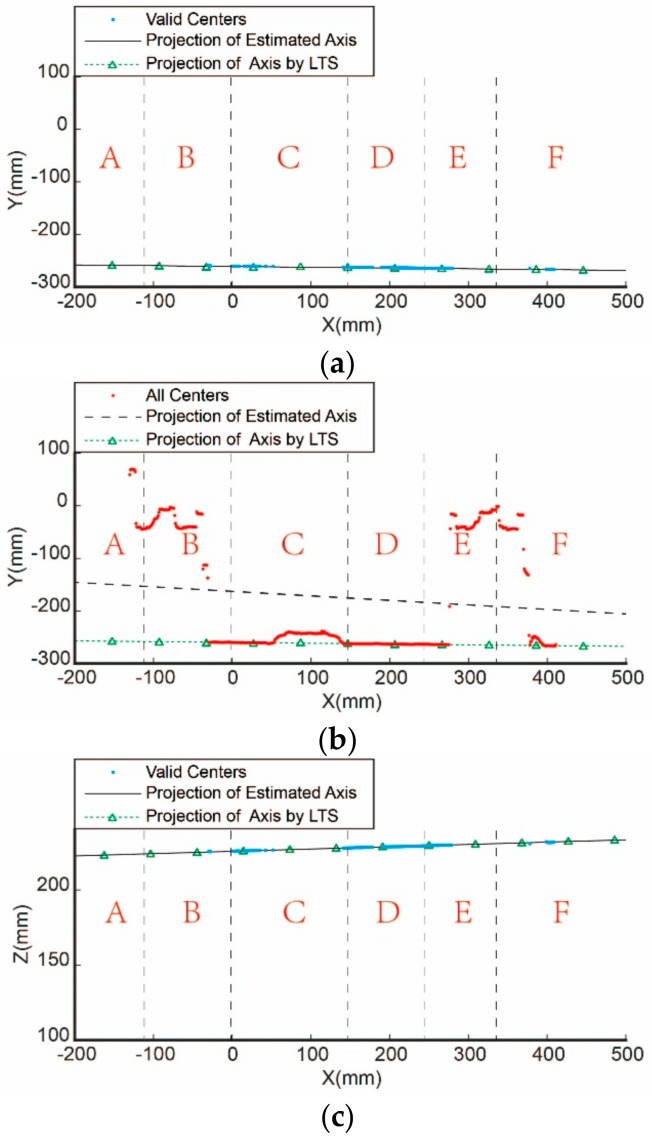
Comparison of the estimated center and axis by the robust method and the non-robust method: (**a**) Projection of the estimated centers and axis on the XOY
by the proposed method; (**b**) projection of the estimated centers and axis on the XOY by the non-robust method; (**c**) projection of the estimated centers and axis on the XOZ by the proposed method; (**d**) projection of the estimated centers and axis on the XOZ by the non-robust method.

**Figure 11 sensors-19-02234-f011:**
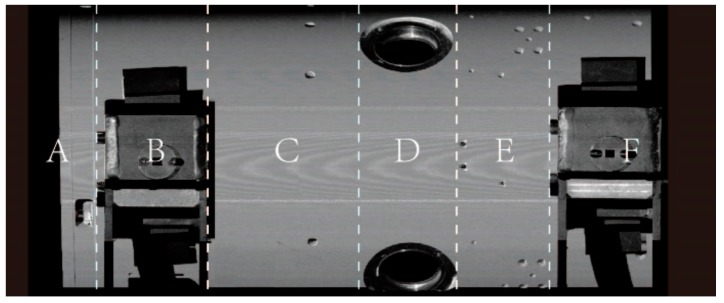
A referenced side view of the cylindrical part.

**Table 1 sensors-19-02234-t001:** The estimated center by general algebraic fitting (mm).

	$X_{C}$	$Y_{C}$	Inner Percent by Measurement	Real Inner Percent	Error
Ideal	0	0	-	-	-
A	−0.0050	−0.0128	97.5%	100%	0.0137
B	3.0625	0.0371	7.6%	96%	3.0627
C	−2.1428	−0.0250	2.1%	85%	2.1429
D	158.5066	4.2871	0.35%	50%	158.5646

**Table 2 sensors-19-02234-t002:** The estimated center by improved M-estimation fitting (mm).

	$X_{C}$	$Y_{C}$	Inner Percent by Measurement	Real Inner Percent	Error
Ideal	0	0	-	-	-
A	−0.0088	−0.0132	97.3%	100%	0.0159
B	0.0367	−0.0002	93.2%	96%	0.0367
C	0.7803	−0.0059	81.3%	85%	0.7803
D	157.6744	4.0763	0.35%	50%	157.7271

**Table 3 sensors-19-02234-t003:** Pose parameter acquired by the proposed method and non-robust method.

Method\Pose	β/°	γ/°	yC0/mm	zC0/mm
Proposed	−0.853	0.826	−0.114	0.845
Non-robust	−4.906	0.430	98.145	−19.732
LTS	−0.868	0.821	0.042	0.613
Deviation A ^1^	0.015	0.005	0.156	0.241
Deviation B ^2^	4.038	0.425	98.103	−19.119

^1^ Deviation between the pose parameters obtained by the proposed method and the LTS. ^2^ Deviation between the pose parameters obtained by the proposed method and the Non-robust method.

**Table 4 sensors-19-02234-t004:** Pose parameters before the adjustment and after the adjustment.

Method\Pose	β/°	γ/°	yC0/mm	zC0/mm
Target pose	0	0	0	0
Pose A ^1^	−0.011	0.022	0.081	−0.021
Pose B ^2^	0.009	0.017	−0.120	0.080
Deviation between the pose A and pose B	0.020	0.005	0.201	0.101

^1^ Pose parameters obtained by laser scanning through the proposed method. ^2^ Pose parameters Pose parameters obtained by the LTS.

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
