# Peer review of "An Improved Robust Method for Pose Estimation of Cylindrical Parts with Interference Features"

_sensors, 2019, doi:10.3390/s19102234_

Round 1
Reviewer 1 Report
In my opinion, the article does not require any significant corrections. The text introduces new scientific solutions (in terms of its real-world applications).
My remarks concern a too modest review of the literature in Introduction. Please try to refer to the texts in the discussion:
- DOI: 10.24425/118156
- DOI: 10.1051/e3sconf/20182600014
- DOI: 10.1061/(ASCE)SU.1943-5428.0000183
Please also look at Table 1. Some numbers have a different number of digits after the decimal point. In the engineering record it proves the different accuracy of the input data and the different accuracy of the results. The Bradis-Krylov rule applies here. Please check and possibly correct the entry (the entire column of Real Innert Percent and Error: 158.56). Similarly in table 2.
Author Response
Thanks for your kindly work and we have learned much from the comments given by you. In the revised manuscript, we use the “Track Changes” function to underline the changes in Microsoft Word.
The changes are listed as follows according to your comments:
1. The suggested texts are introduced as references [32], [33] and [34] in Ln: 218, 272 and 275.
2. The number of the digits after the decimal point in Table 1 and 2 has been adjusted according to the Brand-Krylov rules.
Reviewer 2 Report
This is a well-written and well-presented paper
The authors understand 3d point cloud data and its application (cross section data), which is nicely presented
The methodology is proved with prototype experiment
The paper is prepared according MDPI author instructions. All images are very good, they have a proper caption, on right place in the paper, easy to understand - nice work
Introduction is well written, it gives the proper literature review, please consider to add the sources which you have probably used (examples: ln 40-42; 63-65, 71-72, 83, …. is missing literature source)
Ln: 192-203 ????? some parts of the MDPI author instructions are in the paper, please recheck the paper
Use the same term for Figs. and Figure, (ln. 329 use Figures as in the rest of the paper)
Maybe, discuss few sentences more about the possible application of presented method in practical engineering.
Author Response
Thanks for your kindly work and we have learned much from your comments. In the revised manuscript, we use the “Track Changes” function to underline the changes in Microsoft Word.
The changes are listed as follows:
1. Literature sources of references [2], [8] and [14] are appended in Ln: 41-43, 64-66, 72-73 and 84.
2. Some part of the MDPI author instructions are included in Ln: 193-204 due to our carelessness, now they have been deleted. We are terribly sorry for that.
3. Figs. in Ln: 332 is changed to Figure.
4. Some descriptions of possible applications are attached at the end of the manuscript in Ln: 485-490.
5. All the numbers of the references are adjusted for the addition of the references.